# Multi-Neuron Unleashes Expressivity of ReLU Networks Under Convex Relaxation

## Abstract

Neural work certification has established itself as a crucial tool for ensuring the robustness of neural networks. Certification methods typically rely on convex relaxations of the feasible output set to provide sound bounds. However, complete certification requires exact bounds, which strongly limits the expressivity of ReLU networks: even for the simple "max" function in $\mathbb{R}^2$, there does not exist a ReLU network that expresses this function and can be exactly bounded by single-neuron relaxation methods. This raises the question whether there exists a convex relaxation that can provide exact bounds for general continuous piecewise linear functions in $\mathbb{R}^n$. In this work, we answer this question affirmatively by showing that (layer-wise) multi-neuron relaxation provides complete certification for general ReLU networks. Based on this novel result, we show that the expressivity of ReLU networks is no longer limited under multi-neuron relaxation. To the best of our knowledge, this is the first positive result on the completeness of convex relaxations, shedding light on the practice of certified robustness.

## 1 Introduction

Neural networks have been shown vulnerable to adversarial attacks (Szegedy et al., 2014), where a small perturbation to the input can lead to a misclassification. The area of adversarial robustness, which measures the robustness of a model with respect to adversarial perturbations, has received much research attention in recent years, reflecting a major concern in the application of neural networks, especially in safety-critical domains such as autonomous driving and medical diagnosis. However, computing the exact adversarial robustness of a neural network is generally NP-hard (Katz et al., 2017), while adversarial attacks which try to construct an adversarial perturbation can only provide an upper bound on the robustness of the model. To tackle this issue, neural network certification (Singh et al., 2018; Wang et al., 2018; Bunel et al., 2020) has been proposed to provide robustness guarantees. Complete certification methods (Katz et al., 2017; Tjeng et al., 2019) that can provide exact bounds for all ReLU networks are computationally expensive due to the inherent hardness of the problem, and thus incomplete methods (Wong & Kolter, 2018; Singh et al., 2018; Weng et al., 2018; Gehr et al., 2018; Xu et al., 2020) have been widely investigated, typically focusing on convex relaxations, which can provide efficient and scalable certification at the cost of losing precision. Beyond certification, all existing algorithms for training certifiable models (Shi et al., 2021; Müller et al., 2023; Mao et al., 2023; 2024a; Palma et al., 2023; Balauca et al., 2024) are also based on convex relaxations. Due to their central role in certified robustness, it is critical to understand the trade-off between the efficiency and precision of convex relaxations.

**Expressivity of ReLU networks under convex relaxations** In this work we focus on studying the expressivity of ReLU networks when convex relaxations are used. It has been previously shown that ReLU networks are expressive: they can precisely express every continuous piecewise linear function (Hertrich et al., 2021) and thus can approximate every continuous function within an arbitrary error rate. A key question here is: do existing convex relaxation methods limit this expressive power? The latest research results suggest a nuanced answer. Interval Bound Propagation (IBP) applies the least precise single-neuron interval relaxation to each neuron, but for every continuous function in $\mathbb{R}^n$ and an arbitrarily small error rate $\delta$, there exists a ReLU network that approximates the function with error $\delta$ and the relaxation error of IBP for this network is less than $\delta$ (Baader et al., 2020). However, Mirman et al. (2022) show that IBP cannot provide exact bounds for gen-

eral continuous piecewise linear functions. Baader et al. (2024) further show that the most precise single-neuron convex relaxation (strictly more precise than IBP), namely the Triangle relaxation (Wong & Kolter, 2018), cannot provide exact bounds for any ReLU network that expresses the "max" function on a compact domain in $\mathbb{R}^2$. This is the case even though the function can be easily expressed without error by a ReLU network with only two ReLU neurons. These results raise the question of whether there exists a convex relaxation $\mathbb{P}$ that does not limit the expressive power of ReLU networks. Concretely:

*Given an arbitrary continuous piecewise linear function in $\mathbb{R}^n$ with a compact domain, can we find a ReLU network that expresses this function such that applying $\mathbb{P}$ to the network returns the function's range exactly?*

**This work: multi-neuron relaxations do not restrict the expressive power of ReLU networks** In this work we address the above question and show that in fact a multi-neuron relaxation which computes the convex hull of input and output variables layer-wise is complete for general (feedforward and skip-connected) ReLU networks with cost related to the number of unstable neurons per layer. When limited to computing the convex hull of only output variables layer-wise, the relaxation is still complete for feedforward networks with cost relying on the number of neurons per layer (network width). Based on these novel results, we show that a multi-neuron relaxation can precisely express every continuous piecewise linear function in $\mathbb{R}^n$, in sharp contrast to any single-neuron relaxation. To the best of our knowledge, this is the first positive result on the completeness of convex relaxations and their expressiveness for continuous piecewise linear functions in high dimensions, leading to a deeper understanding of convex relaxations and their application to certified robustness.

## 2 RELATED WORK

We now briefly review related work most closely related to ours.

**Neural Network Certification.** Existing methods for neural network certification can be categorized into complete methods and incomplete methods. Complete methods provide exact bounds for the output of a network, usually relying on solving a mixed-integer program (Tjeng et al., 2019) or a satisfiability modulo theory problem (Katz et al., 2017). These methods are naturally computationally expensive and do not scale well. Incomplete methods, on the other hand, provide sound but inexact bounds, based on convex relaxations of the feasible output set of a network. Xu et al. (2020) characterizes widely-recognized convex relaxations (Mirman et al., 2018; Wong et al., 2018; Zhang et al., 2018; 2022; Ferrari et al., 2022) as linear constraints, equivalent to linear programming in the corresponding linear systems. We distinguish three concrete convex relaxation methods typically considered by theoretical work: Interval Bound Propagation (IBP) (Mirman et al., 2018; Gowal et al., 2018), which ignores the interdependency between neurons and use interval $\{[a, b] \mid a, b \in \mathbb{R}\}$ as the convex relaxation; Triangle relaxation (Wong & Kolter, 2018), which approximates the ReLU function by a triangle in the input-output space; and multi-neuron relaxations (Singh et al., 2018) which considers a group of ReLU neurons jointly in the linear system.

**Convex Relaxation Theories.** Baader et al. (2020) first show the universal approximation theorem for certified models, stating that for every continuous piecewise linear function $f : \mathbb{R}^n \to \mathbb{R}$ and any error rate $\epsilon > 0$, there exists a ReLU network that expresses $f$ and its IBP analysis can provide bounds with error at most $\epsilon$. This result is generalized to other activations by Wang et al. (2022). However, Mirman et al. (2022) show that there exists a continuous piecewise linear function for which IBP analysis of any finite ReLU network expressing this function cannot provide exact bounds. This means that even for continuous piecewise linear functions, IBP requires a network with infinitely many parameters to provide exact bounds. Further, Mao et al. (2024b) show that IBP introduces a strong regularization on the parameter signs to provide good bounds, severely limiting the network capability. Beyond IBP, Baader et al. (2024) show that even Triangle, the most precise single-neuron relaxation, cannot precisely express the "max" function in $\mathbb{R}^2$ with a finite ReLU network, although it can precisely express more functions than IBP in $\mathbb{R}$. In sharp contrast, this work shows that a multi-neuron relaxation can precisely express every continuous piecewise linear function in $\mathbb{R}^n$ with a finite ReLU network, providing a positive result on the expressiveness of convex relaxations for continuous piecewise linear functions in high dimensions.

## 3 BACKGROUND

We now start with a brief review of the required background. We first introduce convex relaxations for network certification and then present single-neuron and multi-neuron relaxation methods.

**Convex Relaxations for Certification.** Given a function $f : \mathbb{R}^{d_{\text{in}}} \to \mathbb{R}^{d_{\text{out}}}$ and a compact domain $X \subset \mathbb{R}^{d_{\text{in}}}$, we denote the graph of the function $\{(\boldsymbol{x}, f(\boldsymbol{x})) \in \mathbb{R}^{d_{\text{in}} + d_{\text{out}}} : x \in X\}$ by $f[X]$. The certification task boils down to computing the upper and lower bounds of the range $f(X)$, in order to verify that these bounds meet certain requirements, e.g., adversarial robustness. To this end, convex relaxations approximate $f[X]$ by a convex set $S \subset \mathbb{R}^{d_{\text{in}} + d_{\text{out}}}$ satisfying $S \supseteq f[X]$. We then take the upper and lower bounds of $S$ (projected into $\mathbb{R}^{d_{\text{out}}}$)—which are usually much easier to compute compared to those of $f(X)$

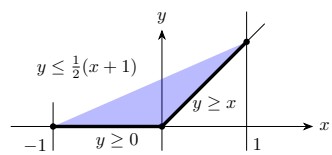

Figure 1: Triangle relaxation of a ReLU with input $x \in [-1, 1]$.

due to the convexity of $S$—as an over-approximation of the bounds on $f(X)$. We assume the domain $X$ to be a convex polytope, because this is the common practice in certification, e.g., $L_\infty$ neighborhoods of a reference point. Such convex polytopes can be represented by a set of linear constraints $\mathcal{C}(\boldsymbol{x}, f(\boldsymbol{x})) \leq \boldsymbol{0}$. For example, consider the ReLU function $y = \max(0, x)$ on the domain $X = [-1, 1]$. One possible convex relaxation is the Triangle relaxation (Wong & Kolter, 2018), represented by the set of linear constraints $(y \geq x) \wedge (y \geq 0) \wedge \left[ y \leq \frac{1}{2}(x+1) \right]$. Figure 1 illustrates this, where the black thick line represents $f[X]$ and the colored area stands for $S$.

**ReLU Network Analysis with Layer-wise Convex Relaxations.** Computing $f[X]$ of a ReLU network is generally NP-hard. To ease the computation, convex relaxations are applied in a layer-wise manner. Specifically, consider a ReLU network $f = \boldsymbol{W}_L \circ \rho \circ \cdots \circ \rho \circ \boldsymbol{W}_1$ and an input convex polytope $X$. Denote the variable of the input layer by $\boldsymbol{x}^{(0)}$, the first layer by $\boldsymbol{x}^{(1)} = \boldsymbol{W}_1(\boldsymbol{x}^{(0)})$, the second layer by $\boldsymbol{x}^{(2)} = \rho(\boldsymbol{x}^{(1)})$, and so on. Assume the input polytope is defined by the linear constraint set $\mathcal{C}_0(\boldsymbol{x}^{(0)}) \leq \boldsymbol{0}$. We apply convex relaxations to the first layer $\boldsymbol{x}^{(1)} = \boldsymbol{W}_1(\boldsymbol{x}^{(0)})$ to obtain a set of linear constraints $\mathcal{C}_1(\boldsymbol{x}^{(0)}, \boldsymbol{x}^{(1)}) \leq \boldsymbol{0}$. Proceeding by layers, we obtain linear constraint sets $\mathcal{C}_{\ell+1}(\boldsymbol{x}^{(\ell)}, \boldsymbol{x}^{(\ell+1)}) \leq \boldsymbol{0}$, for $\ell = 0, \ldots, 2L-2$. Note that no explicit constraint across layers is considered, e.g., $\mathcal{C}(\boldsymbol{x}^{(0)}, \boldsymbol{x}^{(2L-1)}) \leq \boldsymbol{0}$ would not appear explicitly in the above procedure. Finally, we take the union of all constraint sets, $\mathcal{C} = \mathcal{C}_0(\boldsymbol{x}^{(0)}) \cup \mathcal{C}_1(\boldsymbol{x}^{(0)}, \boldsymbol{x}^{(1)}) \cup \cdots \cup \mathcal{C}_{2L-1}(\boldsymbol{x}^{(2L-2)}, \boldsymbol{x}^{(2L-1)})$ and solve $\mathcal{C} \leq \boldsymbol{0}$ by by linear programming to obtain the upper and lower bounds of the output variable $\boldsymbol{x}^{(2L-1)}$. As we perform the relaxation on $\boldsymbol{W}_\ell(\cdot)$ or $\rho(\cdot)$ for every layer, the set $\mathcal{C}$ represents a convex relaxation of the overall composed function $f = \boldsymbol{W}_L \circ \rho \circ \cdots \circ \rho \circ \boldsymbol{W}_1$ on domain $X$. Note that we can choose to further neglect part of the linear constraints to reduce the computational complexity, yielding a more loose relaxation.

**Single-Neuron and Multi-Neuron Relaxations.** Within the framework of layer-wise convex relaxations, the constraint set of an affine layer $\boldsymbol{y} = \boldsymbol{A}\boldsymbol{x} + \boldsymbol{b}$ is always $\mathcal{C}(\boldsymbol{x}, \boldsymbol{y}) = \{\boldsymbol{A}\boldsymbol{x} + \boldsymbol{b} - \boldsymbol{y}, -\boldsymbol{A}\boldsymbol{x} - \boldsymbol{b} + \boldsymbol{y}\} \leq \boldsymbol{0}$, which translates to the equality $\boldsymbol{y} = \boldsymbol{A}\boldsymbol{x} + \boldsymbol{b}$. No loss of precision, therefore, is introduced in affine layers. The core difference between different relaxation methods is how they handle the ReLU function. Single-neuron relaxation methods relax each ReLU neuron separately and disregard the interdependence between neurons, while multi-neuron relaxations consider a group of ReLU neurons jointly. Concretely, for the ReLU layer $\boldsymbol{y} = \rho(\boldsymbol{x})$ with $\boldsymbol{x} \in \mathbb{R}^d$, the constraint sets computed by single-neuron relaxations are of the form $\mathcal{C}(\boldsymbol{x}_i, \boldsymbol{y}_i)$ with $i \in [d]$. In contrast, multi-neuron relaxations produce constraint sets of the form $\mathcal{C}(\boldsymbol{x}_{I_1}, \boldsymbol{y}_{I_2})$ with $I_1, I_2 \subseteq [d]$.

Singh et al. (2019) propose the first multi-neuron relaxation called $k$-ReLU. For each ReLU layer, it considers at most $k$ unstable neurons jointly, i.e., $\mathcal{C}(\boldsymbol{x}, \boldsymbol{y})$ is of the form $\mathcal{C}(\boldsymbol{x}_I, \boldsymbol{y}_I)$, with $I \subseteq [d], |I| \leq k$. However, $k$-ReLU is not complete for general ReLU networks (see §7), thus we consider a stronger multi-neuron relaxation which only restrict the number of output variables in the constraints, allowing $\mathcal{C}(\boldsymbol{x}, \boldsymbol{y})$ to be of the form $\mathcal{C}(\boldsymbol{x}, \boldsymbol{y}_I)$ with $I \subseteq [d], |I| \leq k$. We denote this special multi-neuron relaxation as $\mathbb{M}_k$, and assume it always computes the convex hull of $(\boldsymbol{x}, \rho(\boldsymbol{s}_I))$ while only one index set $I$ is allowed per layer for simplicity. We also consider the weaker output-only multi-neuron relaxation $\mathbb{M}_k^o$ which only computes the convex hull of the output set. Concretely, $\mathcal{C}(\boldsymbol{x}, \rho(\boldsymbol{x}))$ is in the form of $\{\mathcal{C}(\rho(\boldsymbol{x}_I)) \mid I \subseteq [d], |I| \leq k\}$, and only one index set $I$ is allowed per layer as well. For $\mathbb{M}_k^o$, we will not solve the full system but only take the constraints computed for

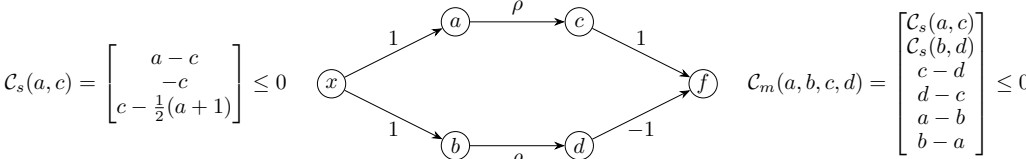

Figure 2: Visualization of the single-neuron and multi-neuron relaxations for a network encoding $f(x) = 0$.

the last layer, and denote the convex polytope defined by the constraints computed for the last layer as $\mathbb{M}_k^o(f, X_0)$. Intuitively, $\mathbb{M}_k^o$ relaxes the functional range, while $\mathbb{M}_k$ relaxes the functional graph (domain and range) jointly. We note that $\mathbb{M}_k^o$ is allowed to consider unstable and stable neurons together, while $k$-ReLU only considers unstable neurons together with the corresponding inputs, thus they are not comparable in precision even when $k$-ReLU also computes the convex hull of the considered variables. Neurons that are not considered by a multi-neuron relaxation are processed by a single-neuron relaxation, in our case, the Triangle relaxation. We remark that there are other applied multi-neuron relaxations (Müller et al., 2022; Ferrari et al., 2022) that only compute an over-relaxation of $\mathbb{M}_k$.

**Numerical Illustration.** We include a toy example to illustrate the concepts introduced above, namely the ReLU network $\rho(x) - \rho(x)$ encoding the zero function $f(x) = 0$ with input $x \in [-1, 1]$. This network is visualized in Figure 2. The linear constraints are as follows: (i) for the input convex polytope, we have $(x - 1 \le 0) \wedge (-1 - x \le 0)$; (ii) for affine layers, we have $(a = x) \wedge (b = x) \wedge (f = c - d)$; (iii) for the ReLU layer, a single neuron relaxation (Triangle) will have $[\mathcal{C}_s(a, c) \le 0] \wedge [\mathcal{C}_s(b, d) \le 0]$, and a multi-neuron relaxation ($\mathbb{M}_2$) will have $\mathcal{C}_m(a, b, c, d) \le 0$. In this case, a multi-neuron relaxation successfully solves that the upper bound and lower bound of $f$ are zero, while a single-neuron relaxation solves that the upper bound is 1 and the lower bound is -1 which are not exact. In addition, the output-only multi-neuron relaxation $\mathbb{M}_2^o$ first computes the output convex polytope relaxation of the first layer $(a = b) \wedge (1 - a \le 0) \wedge (a - 1 \le 0)$, and then computes the output convex polytope relaxation of the second layer given the previous polytope, which is $(c = d) \wedge (c \ge 0) \wedge (c \le 1)$. Proceeding layer-wisely, we obtain the final convex polytope $f = 0$, thus the bounds from $\mathbb{M}_2^o$ are also exact. Note that $\mathbb{M}_k$ is solved with linear programming on the induced constraints for all layers, while $\mathbb{M}_k^o$ is solved only for the last layer, i.e., it finds the maximum and minimum in the final convex polytope.

## 4 FULL EXPRESSIVITY OF RELU NETWORK UNDER MULTI-NEURON RELAXATIONS

We now present our main result. We combine an existing result on the representation capability of ReLU networks with our novel results, which we prove in detail in §5 and §6, to answer the question posed in §1.

We establish in §5 that $\mathbb{M}_k^o$ returns exact bounds for ReLU networks of width no more than $k$. In §6, we prove that if a ReLU network has at most $k$ unstable neurons in each layer—this number could be far smaller than the network width—then $\mathbb{M}_k$ provides exact output bounds. As a final step towards Theorem 1 below, Lemma 1 (Hanin, 2019, Theorem 2) states that any continuous piecewise linear function $f : [0, 1]^{d_{\text{in}}} \to \mathbb{R}$ can be expressed by a ReLU network of width $d_{\text{in}} + 3$ which has at most 3 unstable neurons per layer.

**Theorem 1.** *Let $d_{in} \in \mathbb{N}$ and let $X \subseteq [0, 1]^{d_{in}}$ be a convex polytope in $\mathbb{R}^{d_{in}}$. For every continuous piecewise linear function $f : [0, 1]^{d_{in}} \to \mathbb{R}$, denote the lower and upper bound of the range $f(X)$ by $l := \min_{x \in X} f(x)$ and $u := \max_{x \in X} f(x)$. Then there exists a ReLU network $\Phi$ satisfying $\Phi(x) = f(x), \forall x \in X$, and applying $\mathbb{M}_3$ and $\mathbb{M}_{d_{in}+3}^o$ to $(\Phi, X)$ both return $l$ and $u$.*

*Proof.* By Lemma 1 below, there exists a ReLU network $\Phi$ of width $d_{\text{in}} + 3$ with at most 3 unstable neurons per layer satisfying $\Phi(x) = f(x)$, for $x \in X$. Theorem 2 in §5 shows that applying $\mathbb{M}_{d_{\text{in}}+3}^o$ to $(\Phi, X)$ returns the exact upper and lower bounds of $\Phi$ on $X$; Theorem 6 in §6 shows that applying

$\mathbb{M}_3$ to $(\Phi, X)$ returns the exact upper and lower bounds of $\Phi$ on $X$. Since $\Phi = f$ on $X$, the bounds of $\Phi$ coincides with those of $f$. This establishes the claim. $\quad\square$

**Lemma 1.** (Hanin, 2019, Theorem 2) Let $d_{\text{in}} \in \mathbb{N}$ and let $f : [0,1]^{d_{\text{in}}} \to \mathbb{R}$ be a continuous piecewise linear function. There exists a ReLU network $\Phi$ of width $d_{\text{in}} + 3$ and finite depth, satisfying

$$\Phi(x) = f(x), \text{ for } x \in [0, 1].$$

Furthermore, $\Phi$ has at most 3 unstable neurons in each layer.

*Proof.* We refer to (Hanin, 2019, Theorem 2) for the constructive proof. We only note that in each hidden layer of the constructed network, $d_{\text{in}}$ neurons are copies of the input variables. Thus the network has at most 3 unstable neurons per layer. $\quad\square$

## 5 MULTI-NEURON EXPRESSIVITY WITH BOUNDED WIDTH

We now develop the first central result behind our main theorem on the expressivity, which shows that the output-only multi-neuron relaxation $\mathbb{M}_k^o$ introduced in §3 solves the exact output bound for ReLU networks of width at most $k$. This result is formally presented in Theorem 2.

**Theorem 2** (Precise $\mathbb{M}_k^o$ with Bounded Width). Let $L, k, d_{\text{in}}, d_{\text{out}} \in \mathbb{N}$. Consider a ReLU network $f : \mathbb{R}^{d_{\text{in}}} \to \mathbb{R}^{d_{\text{out}}}$ of depth $L$ and width $\leq k$. Let $X \subset \mathbb{R}^{d_{\text{in}}}$ be a convex polytope. Applying $\mathbb{M}_k^o$ to $f$ on domain $X$ returns the exact output set which is also a convex polytope, i.e.,

$$\mathbb{M}_k^o(f, X) = f(X). \tag{1}$$

*Proof.* We prove by induction on the network depth $L$ that $\mathbb{M}_k^o(f, X) = f(X)$. By Lemma 3 below, $f(X)$ is a convex polytope for every ReLU network $f$.

We start with the base case $L = 1$, when $f$ is an affine function $f(\boldsymbol{x}) = \boldsymbol{A}\boldsymbol{x} + \boldsymbol{b}$. By definition, $\mathbb{M}_k^o(f, X) = \{\boldsymbol{A}\boldsymbol{x} + \boldsymbol{b} \mid \boldsymbol{x} \in X\} = f(X)$. To prove the induction step, we assume that (1) holds for all ReLU networks of depth $\leq L - 1$ and width $\leq k$. The subnetwork $f' = \boldsymbol{W}_{L-1} \circ \rho \circ \cdots \circ \boldsymbol{W}_1$ consisting of the first $L - 1$ affine and ReLU layers of $f$, clearly, has depth $L - 1$ and width $\leq k$. By induction hypothesis, $\mathbb{M}_k^o(f', X) = f'(X)$. The resting subnetwork $f'' = \boldsymbol{W}_L \circ \rho$ which consists of the last affine and ReLU layer of $f$, or equivalently $f'' = \boldsymbol{W}_L \circ \rho \circ \text{Identity}$, has depth 2 and width $\leq k$. By induction hypothesis, again, we have $\mathbb{M}_k^o(f'', f'(X)) = f''(f'(X))$. Therefore,

$$\mathbb{M}_k^o(f, X) = \mathbb{M}_k^o(f'' \circ f', X) = \mathbb{M}_k^o(f'', f'(X)) = f''(f'(X)) = f(X).$$

This concludes the proof of the induction step and hence establishes the claim. $\quad\square$

Theorem 2 is mainly based on two observations. First, the convex hull of a convex polytope is the polytope itself; in other words, $\mathbb{M}_k^o$ does not introduce any relaxation error for a single layer when the feasible output set under consideration is a convex polytope, as illustrated in Figure 3. Second, ReLU networks transform convex polytopes into convex polytopes, as illustrated in Figure 4. This convex polytope preserving property is proved in Lemma 3.

**Lemma 3.** Let $d_{\text{in}}, d_{\text{out}} \in \mathbb{N}$, $f : \mathbb{R}^{d_{\text{in}}} \to \mathbb{R}^{d_{\text{out}}}$ be a ReLU network ended with either affine or ReLU layer, and $X$ be a convex polytope in $\mathbb{R}^{d_{\text{in}}}$. Then $f(X)$ is a convex polytope in $\mathbb{R}^{d_{\text{out}}}$.

*Proof.* We first show that every affine and ReLU layer transforms a convex polytope into a convex polytope. Then, we prove the statement by induction on the network depth.

Assume the input convex polytope $X$ is represented by linear constraint set $\mathcal{C}(\boldsymbol{x}) \leq \boldsymbol{0}$. Consider an affine transformation $\boldsymbol{y} = \boldsymbol{A}\boldsymbol{x} + \boldsymbol{b}$. The functional graph $\{(\boldsymbol{x}, \boldsymbol{y}) : \boldsymbol{y} = \boldsymbol{A}\boldsymbol{x} + \boldsymbol{b}, \boldsymbol{x} \in X\}$ is defined by the constraints $\{\mathcal{C}(\boldsymbol{x}), \boldsymbol{y} - \boldsymbol{A}\boldsymbol{x} - \boldsymbol{b}, -\boldsymbol{y} + \boldsymbol{A}\boldsymbol{x} + \boldsymbol{b}\} \leq \boldsymbol{0}$. Eliminating the variable $\boldsymbol{x}$ using the Fourier–Motzkin algorithm (Fourier, 1827), the resulting constraints are affine inequalities of $\boldsymbol{y}$, thus define a convex polytope for $\boldsymbol{y}$. We proceed to show the same property holds for the ReLU function $\boldsymbol{y} = \rho(\boldsymbol{x})$. Assume again that the input convex polytope $X$ is represented by linear constraint set $\mathcal{C}(\boldsymbol{x}) \leq \boldsymbol{0}$. The range of this function is then represented by the constraints $\{\mathcal{C}(\boldsymbol{y}), -\boldsymbol{y}\} \leq \boldsymbol{0}$, which defines a convex polytope.

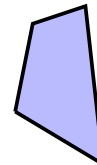
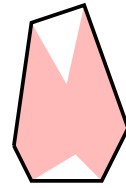

Figure 3: $\mathbb{M}^o$ returns the convex hull of output set (black thick boundaries). When the output set is a convex polytope (left, shaded blue), $\mathbb{M}^o$ returns the exact output set. When the output set is not a convex polytope (right, shaded red), $\mathbb{M}^o$ introduces imprecision.

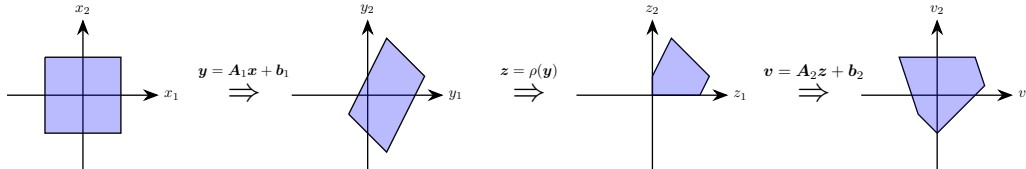

Figure 4: A convex polytope in $\mathbb{R}^2$ retains as a convex polytope under arbitrary compositions of affine and ReLU transformations.

Now we prove the claim by induction on the network depth $L$. The base case $L = 1$ directly follows from the convex polytope preserving property of affine transformations we established above. For the induction step, we assume that ReLU networks of depth $L-1$ transforms a convex polytope into a convex polytope. The subnetwork $f' = \boldsymbol{W}_{L-1} \circ \rho \circ \cdots \circ \boldsymbol{W}_1$ has depth $L-1$ and, by the induction hypothesis, transforms $X$ into the convex polytope $f'(X)$. The resting subnetwork $f'' = \boldsymbol{W}_L \circ \rho$ has depth 2 and thus by the induction hypothesis transforms $f'(X)$ into a convex polytope. This completes the induction step and concludes the proof of the lemma. $\qquad\square$

## 6    MULTI-NEURON EXPRESSIVITY WITH BOUNDED UNSTABLE NEURONS

We have shown in §5 that the output-only multi-neuron relaxation $\mathbb{M}^o_k$ returns the exact output set for ReLU networks of width at most $k$. This result essentially relies on the fact that in a feedforward ReLU network, $\mathbb{M}^o_k$ does not lose precision for layers with at most $k$ neurons, although it discards the dependency between input variable and output variable in each layer after processing. However, this result does not directly apply to ReLU networks with skip connections, where neurons between non-adjacent layers might be connected by a skip-connection. While it is also possible to convert a ReLU network with skip connections into a feedforward network by introducing additional neurons in those layers, the width of the resulting feedforward network becomes unnecessarily large, thus $k$ also needs to be as large which leads to significant computational overhead.

In this section, we tackle this problem by developing a general result with $\mathbb{M}_k$ that applies to all ReLU networks, including those with skip connections. Specifically, we show that $\mathbb{M}_k$ is precise for ReLU networks with at most $k$ unstable neurons in each hidden layer. Since the number of unstable neurons in each layer will not increase when converting a network with skip connections to a feedforward network, this result generalizes to ReLU networks with skip connections as well.

We begin by formally defining stable and unstable neurons in Definition 4 and 5. Intuitively, intrinsically unstable neurons are those that switch their activation pattern in the input set, while bounded unstable neurons are those that are not guaranteed to be stable by a convex relaxation, i.e., they have a positive upper bound and a negative lower bound under the given relaxation.

**Definition 4** (Intrinsically Stable and Unstable Neuron). For a ReLU network $\Phi$ and an input set $X$, a ReLU neuron is called intrinsically unstable on $X$ if there exists $x_1, x_2 \in X$ such that $x_1$ activates this neuron and $x_2$ does not activate it. Otherwise, it is called intrinsically stable on $X$.

**Definition 5** (Bounded Stable and Unstable Neuron). Consider a ReLU network $f = \boldsymbol{W} \circ \rho \circ f'$, where $f'$ is a ReLU network with output dimension $d$. Let $X$ be the input set and $\mathbb{P}$ be a convex relaxation. For each neuron in the layer $\rho$, we call it bounded unstable on $X$ w.r.t. $\mathbb{P}$ if the resulting

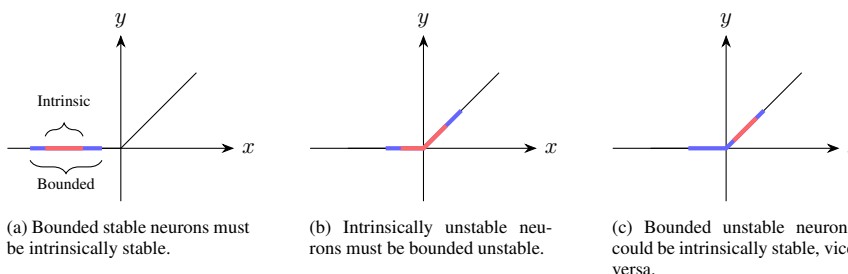

Figure 5: Relationship between intrinsic stability and bounded stability of ReLU neurons.

upper bound on $f'(X)$ obtained by $\mathbb{P}$ is positive and the lower bound is negative. Otherwise, it is called bounded stable w.r.t. $\mathbb{P}$.

We remark that an intrinsically unstable neuron is always bounded unstable w.r.t. any convex relaxation, but the converse is not necessarily true. On the other hand, bounded stable neurons must be intrinsically stable, while intrinsically stable neurons could be bounded unstable due to loss of precision caused by the relaxation. We illustrate this in Figure 5.

If a ReLU neuron is bounded stable w.r.t. a relaxation $\mathbb{P}$, then it reduces to an affine transformation as they have a fixed activation pattern—either equal to the identity function or the zero function. Therefore, bounded stable neurons are processed by convex relaxation methods in the same way as an affine function, by replacing the corresponding ReLU with an identity or a zero function.

We are now ready to present the central result of this section, which states that $\mathbb{M}_k$ is precise for ReLU networks with at most $k$ unstable neurons in each hidden layer.

**Theorem 6** (Precise $\mathbb{M}_k$ with Limited Unstable Neurons). Let domain $X \subset \mathbb{R}^{d_{\text{in}}}$ be a convex polytope. For a ReLU network $f_L \circ f_{L-1} \cdots \circ f_1$ where $f_i$ is an affine layer followed by a ReLU layer except $f_L$ which is a single affine layer, assume $f_i$ has at most $k$ intrinsically unstable neurons for every $i$, linear programming with constraints induced by $\mathbb{M}_k$ on only unstable neurons to $\Phi$ results in exact upper and lower bounds for the final output of the network.

*Proof.* We prove by induction on $L$ that the constraints induced by $\mathbb{M}_k$ have the same feasible set as the constraints induced by $\mathbb{M}_\infty$. Since $\mathbb{M}_\infty$ is more precise than $\mathbb{M}_\infty^o$ and $\mathbb{M}_\infty^o$ returns the exact output set (Theorem 2), this implies Theorem 6. Base case: when $L = 1$, the ReLU network is simply an affine layer, thus constraints induced by both relaxations are $\boldsymbol{u} = \boldsymbol{A}\boldsymbol{x} + \boldsymbol{b}$ where $\boldsymbol{A}\boldsymbol{x} + \boldsymbol{b}$ is the affine layer. Inductive step: assume that $\mathbb{M}_k$ has equivalent constraints as $\mathbb{M}_\infty$ for $f' := f_{L-1} \cdots \circ f_1$. By the induction hypothesis, constraints on $f'$ define the exact output set of $f'$. Thus, since $f_L$ has at most $k$ intrinsically unstable neurons, it has at most $k$ bounded unstable neurons. Therefore, by Lemma 8 (proved later), $\mathbb{M}_k$ for $f$ still has equivalent constraints as $\mathbb{M}_\infty$. $\square$

Theorem 6 relies on the observation (Lemma 8) that the constraints induced by $\mathbb{M}_k$ on the unstable neurons are equivalent to the constraints induced by $\mathbb{M}_\infty$, in the sense that they have the same feasible set. We now prove this fact in a weak form first, which states that for a single affine layer followed by a ReLU layer, the constraints induced by $\mathbb{M}_k$ on the unstable neurons are equivalent to the constraints induced by the convex hull of the composed function. This is formalized in Lemma 7.

**Lemma 7** (The Strong Form of $\mathbb{M}_k$). For an affine layer $\boldsymbol{u} = \boldsymbol{A}\boldsymbol{x} + \boldsymbol{b}$ followed by a ReLU layer $\boldsymbol{v} = \rho(\boldsymbol{u})$ with $k$ bounded unstable neurons, the constraint set induced by $\mathbb{M}_k$ on the $k$ unstable neurons is equivalent to the constraint set induced by $g[X]$ for $g(\boldsymbol{x}) = \rho(\boldsymbol{A}\boldsymbol{x} + \boldsymbol{b})$ given any convex polytope input set $X$.

*Proof.* We denote the constraint set induced by $\mathbb{M}_k$ to be $\mathrm{LS}_1$ and the constraint set induced by $g[X]$ to be $\mathrm{LS}_2$. Since $\mathrm{LS}_2$ is the convex hull, every solution satisfying $\mathrm{LS}_2$ also satisfies $\mathrm{LS}_1$. In the following, we show that every feasible solution satisfying $\mathrm{LS}_1$ also satisfies $\mathrm{LS}_2$, thus establishing the equivalence between feasible sets of $\mathrm{LS}_1$ and $\mathrm{LS}_2$.

Let linear constraints induced by the input convex polytope be $P(\boldsymbol{x}) \leq \boldsymbol{0}$. Without loss of generality, we assume the first $k$ neurons are unstable and the rest $n - k$ neurons are stable where $n$ is the output dimension. Therefore, $\text{LS}_1$ is $[P(\boldsymbol{x}) \leq \boldsymbol{0}] \wedge [\boldsymbol{u} = \boldsymbol{A}\boldsymbol{x} + \boldsymbol{b}] \wedge [\boldsymbol{v}_{k+1:n} = \boldsymbol{W}_\rho \boldsymbol{u}_{k+1:n}] \wedge [\mathcal{C}_1(\boldsymbol{u}, \boldsymbol{v}_{1:k}) \leq \boldsymbol{0}]$, and $\text{LS}_2$ is $[P(\boldsymbol{x}) \leq \boldsymbol{0}] \wedge [\boldsymbol{u} = \boldsymbol{A}\boldsymbol{x} + \boldsymbol{b}] \wedge [\mathcal{C}_2(\boldsymbol{x}, \boldsymbol{v}) \leq \boldsymbol{0}]$, where $\boldsymbol{W}_\rho$ is the equivalent affine weight (taking 1 or 0 as elements) of ReLU layer for stable neurons, and $\mathcal{C}_1$ and $\mathcal{C}_2$ are the constraints induced by the convex hull of $\rho$ and $\boldsymbol{g}$, respectively. For stable neurons $\boldsymbol{v}_{k+1:n}$, they are affine functions of $\boldsymbol{x}$, i.e., $\boldsymbol{v}_{k+1:n} = \boldsymbol{W}_\rho (\boldsymbol{A}\boldsymbol{x} + \boldsymbol{b})_{k+1:n}$, which is the tightest possible constraints in $\text{LS}_2$ for them, thus every feasible solution $\boldsymbol{v}_{k+1:n}$ for $\text{LS}_1$ also satisfies $\text{LS}_2$ because $\text{LS}_1$ imposes this constraint. Now we consider unstable neurons $\boldsymbol{v}_{1:k} = \rho(\boldsymbol{u}_{1:k})$ where $\boldsymbol{u} = \boldsymbol{A}\boldsymbol{x} + \boldsymbol{b}$. $\mathcal{C}_1$ imposes all possible constraints in $\{l_1(\boldsymbol{u}, \boldsymbol{v}_{1:k}) \leq \boldsymbol{0}\}$ and $\mathcal{C}_2$ imposes all possible constraints in $\{l_2(\boldsymbol{x}, \boldsymbol{v}_{1:k}) \leq \boldsymbol{0}\}$, where $l_i$ are some affine expression of the given variables. Therefore, we can rewrite $l_1$ in $\text{LS}_1$ as $l_1(\boldsymbol{A}\boldsymbol{x} + \boldsymbol{b}, \boldsymbol{v}_{1:k})$. Since $\boldsymbol{v}_{1:k} = \rho(\boldsymbol{A}\boldsymbol{x} + \boldsymbol{b})_{1:k}$, all effective $l_2(\boldsymbol{x}, \boldsymbol{v}_{1:k})$ must also be in the form of $l_2(\boldsymbol{A}\boldsymbol{x} + \boldsymbol{b}, \boldsymbol{v}_{1:k})$. Here effective constraints are those that change the feasible set if removed. Since both $\text{LS}_1$ and $\text{LS}_2$ impose all possible constraints in $\{l(\boldsymbol{A}\boldsymbol{x} + \boldsymbol{b}, \boldsymbol{v}_{1:k}) \leq \boldsymbol{0}\}$, every feasible solution satisfying $\text{LS}_1$ also satisfies $\text{LS}_2$. $\quad\square$

We have shown in Lemma 7 that for a single affine layer followed by a ReLU layer, the constraints induced by $\mathbb{M}_k$ on the unstable neurons are equivalent to the constraints induced by the convex hull of the composed function. We now extend this result to the general case of a ReLU network with at most $k$ unstable neurons in each hidden layer in Lemma 8, which completes the proof of Theorem 6.

**Lemma 8.** For a ReLU network $f_L \circ f_{L-1} \cdots \circ f_1$ where $f_i$ is an affine layer followed by a ReLU layer, given the linear constraints computed for $\{f_i \mid i \leq L - 1\}$ and at most $k$ bounded unstable neurons for $f_L$, constraints induced by $\mathbb{M}_k$ on bounded unstable neurons for $f_L$ has the same feasible set as constraints induced by $\mathbb{M}_\infty$ for $f_L$.

*Proof.* We use contradiction to prove the lemma. Let $f_L$ map $X_L$ to $Y_L$. Suppose $\mathbb{M}_k$ is less precise than $\mathbb{M}_\infty$, then there must exist a linear constraint in $\mathbb{M}_\infty$ for $f_L$ in the space $\mathbb{X}_L \times \mathbb{Y}_L$ that reduces the feasible set of constraints induced by $\mathbb{M}_k$. Denote the set of unstable neurons of $f_L$ as $\mathbb{U}$ and the set of stable neurons of $f_L$ as $\mathbb{S}$. Then we can group the variables in this constraint into $X_L$, $\mathbb{U}$ and $\mathbb{S}$. Since neurons in $\mathbb{S}$ are affine expressions of $X_L$, we can replace them with variables in $\mathbb{X}_L$. The original constraint is then a linear constraint only involving variables in $X_L$ and $\mathbb{U}$. However, by Lemma 7, $\mathbb{M}_k$ already computes the convex hull for $(X_L, \mathbb{U})$, thus such a constraint cannot reduce the feasible set of constraints induced by $\mathbb{M}_k$. Therefore, $\mathbb{M}_k$ for all the unstable neurons in the $f_L$ has the same feasible set as applying $\mathbb{M}_\infty$ for the $L$-th layer. $\quad\square$

# 7 CASE STUDY: THE MAX FUNCTION

Baader et al. (2024) prove that there does not exist a ReLU network that can express the "max" function in the compact domain $[0, 1]^2 \subset \mathbb{R}^2$ such that the network outputs can be bounded exactly by single-neuron relaxations. In this section, we take the "max" function in $\mathbb{R}^d$, $d \geq 2$, on domain $[0, 1]^d$, as an example to show that a multi-neuron relaxation easily resolve such impossibility results, as a confirmation of our main result.

First, consider $d = 2$. In this case, we can represent the "max" function with the ReLU network $f = x_2 + \rho(x_1 - x_2)$, illustrated in Figure 6. This network has width equals two (node $c$ and $d$) and maximum unstable neurons per layer equals one (node $c$). We will thus show that $\mathbb{M}_2^o$ and $\mathbb{M}_1$ can return the exact bounds of the functional range, i.e., $[0, 1]$.

The input constraints are $(x_1 \geq 0) \wedge (x_1 \leq 1) \wedge (x_2 \geq 0) \wedge (x_2 \leq 1)$. Besides, the constraints for affine layers are $(a = x_1 - x_2) \wedge (b = x_2) \wedge (f = c + d)$. With these constraints, we can compute the bounds of the output of the first affine layer with linear programming, yielding $a \in [-1, 1]$ and $b \in [0, 1]$. Therefore, $a$ is bounded unstable and $b$ is bounded stable.

We now show that $\mathbb{M}_1$ computes the exact bounds of $f$. For the bounded stable node $b$, the constraint is $d = b$. For the bounded unstable node $c$, the constraint is $(c \geq 0) \wedge (c \geq a) \wedge (c \leq 1 - b)$. Therefore, we have $f = c + d = c + x_2 \geq 0 + x_2 \geq 0$ and $f = c + d = c + x_2 \leq 1 - x_2 + x_2 = 1$. Thus, $\mathbb{M}_1$ returns the exact bounds of the output of the ReLU network, which is $[0, 1]$. We remark that 1-ReLU (which is equivalent to the Triangle relaxation) cannot return the exact upper bound, as its constraint for node $c$ is $(c \geq 0) \wedge (c \geq a) \wedge (c \leq 0.5a + 0.5)$ since it only allow the constraint of

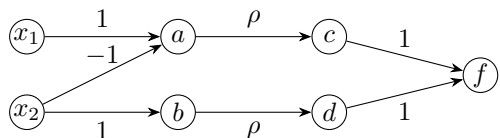

Figure 6: The network encoding $f(x_1, x_2) = \max(x_1, x_2)$.

$c$ to depend on $a$, while $\mathbb{M}_1$ allows it to depend on $b$ as well. Thus, the upper bound of $f$ returned by 1-ReLU is 1.5, which is not exact. This is not surprising because otherwise it will break the results established by Baader et al. (2024).

We further show that $\mathbb{M}_2^o$ also returns the exact bounds for this ReLU network. The input polygon is $(x_1 \in [0, 1]) \wedge (x_2 \in [0, 1])$. Calculating for the first affine layer, the convex polygon returned by $\mathbb{M}_2^o$ is $(a \geq -b) \wedge (a \leq 1 - b) \wedge (b \in [0, 1])$. After the ReLU layer, the convex polygon becomes $(c \geq -d) \wedge (c \leq 1 - d) \wedge (d \in [0, 1]) \wedge (c \geq 0)$. Substituting this into $f = c + d$ and eliminate $c$ and $d$, we get the output convex polygon, $(f \geq 0) \wedge (f \leq 1)$, thus establishes the exact bounds of $f$. We note that in this process, the convex polygon of each layer's output is always exact.

We have shown that a multi-neuron relaxation can exactly bound the network expressing the "max" function in $\mathbb{R}^2$ with the given budget required by Theorem 2 and Theorem 6, respectively. Now we extend the result to $\mathbb{R}^d$. Indeed, we can rewrite "max" in a nested form as $\max(x_1, x_2, \ldots, x_d) = \max(\max(x_1, x_2), \ldots, x_d)$. By the previous argument, a multi-neuron relaxation can bound $u = \max(x_1, x_2)$ exactly. Notice that $u$ has no interdependency with $x_3, \ldots, x_d$, thus we can repeat the procedure for $\max(u, x_3, \ldots, x_d)$. By induction, a multi-neuron relaxation ($\mathbb{M}_2^o$ and $\mathbb{M}_1$) can bound the output of the ReLU network expressing the "max" function in $\mathbb{R}^d$ exactly. We remark that for a "max" function in $\mathbb{R}^d$, two unstable neurons each layer and network width equals $d$ is enough, while Theorem 1 upper bounds this number by 3 and $d + 3$, respectively.

## 8 DISCUSSION

**Complete Certification with Multi-neuron Relaxations.** This work establishes that two particular multi-neuron relaxations are complete verifiers for ReLU networks. Despite their theoretical power, there is currently no algorithmic implementation of these relaxations. In particular, their algorithmic complexity is unknown. Developing efficient algorithms for these relaxations is important for future work, and we suggest a few possible directions below.

The first question is how to compute the convex hull. While this might be exactly computed (e.g., for affine layers), an approximate convex hull might be sufficient for practical purposes (Müller et al., 2022). Therefore, one may rely on "constraint mining", i.e., finding valid constraints sequentially. Since the convex hull is the intersection of all valid constraints, one can iteratively add constraints to the linear system until the convex hull is fully covered. While effective constraint mining is non-trivial, we remark that due to the completeness of multi-neuron relaxations, the expensive branch-and-bound as deployed by Müller et al. (2022) is no longer required to find the exact bounds. In addition, similar constraint mining approaches are deployed by Zhang et al. (2022), but they consider all constraints possibly involving different layers, which is a much larger constraint space than that for a single layer.

The second question is how to solve the linear system efficiently, especially in the process of constraint mining where multiple strongly overlapping linear programming problems need to be solved. This question might be relatively easy, because we can expect the optimal solution of the previous linear programming to be a good initial guess for the next linear programming. In particular, the simplex algorithm might be a good choice for this task because the new optimum must lie on the vertices introduced by new constraints.

The last question is how to check whether we have reached the exact bounds. We suggest two possible approaches. The first approach essentially relies on the effectiveness of constraint mining: if the constraint mining algorithm can no longer find a new constraint that improves the bound, then the current bound is exact. The second approach is to reconstruct the input of the network and check

whether its output matches the current bound. This approach is more straightforward because when solving for $\mathbb{M}_k$, we directly have the values for the input of the network.

**Importance of Certified Training.** Our work shows that ReLU networks with width at most $d+3$ and only three unstable neurons per layer are enough to express any continuous piecewise linear function in $\mathbb{R}^d$, and multi-neuron relaxations can provide exact bounds for these networks. This implies that if we can train customized models, the complexity of certification can be drastically reduced. Therefore, along with more powerful certification tools, the field should develop more powerful training algorithms that can train networks that are easily certifiable.

## 9 CONCLUSION

We proved the first positive result on the completeness of convex relaxations and the expressivity of ReLU networks under convex relaxations. While single-neuron relaxations that relax each neuron separately are incomplete, we proved that (layer-wise) multi-neuron methods, where multiple neurons in the same layer are processed jointly, are complete. Specifically, for networks of width no more than $k$, one computes the convex hull of the range of each layer, proceeding in a layer-wise manner. Then, the resulting set of linear constraints induces exact upper and lower bounds on the output set of the network. In addition, when the network width is unbounded, but the number of unstable neurons is at most $k$ in each layer, we can retain the exact bounds by jointly considering the input-output set of those $k$ neurons. Our results demonstrate that the expressivity of ReLU networks is no longer limited under multi-neuron relaxations, in contrast to single-neuron relaxations which have previously been shown to severely limit the expressivity of networks they can certify exactly.

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
