# OpenReview forum: "Multi-Neuron Unleashes Expressivity of ReLU Networks Under Convex Relaxation"
_ICLR.cc/2025/Conference — ICLR 2025 Conference Withdrawn Submission_

### Official Review · Reviewer_6s15 · 2024-11-01

**Soundness:** 1
**Presentation:** 2
**Contribution:** 2
**Rating:** 1
**Confidence:** 4

**Summary:**

The paper claims that a layerwise (rather than neuronwise) convex relaxation does not limit the expressivity of CWPL functions with ReLU neural networks. More specifically, the authors claim that two relaxations, denoted $M_3$ and $M_{d_{in}+3}^o$, compute the exact lower and upper bounds for a class of neural networks (those with width bounded by the input dimension +3). Since Hanin showed that this class of networks can compute any CPWL function, the relaxations $M_3$ and $M_{d_{in}+3}^o$ are claimed to be complete verifiers in the sense that, for any CPWL function, there exists a neural network such that applying the relaxation yields the exact bounds for the function’s range. Finally, there is a case study of the max functions showing that the multi-neuron convex relaxation can compute exact bounds for the function's range where the single neuron relaxation fails.

**Strengths:**

The paper presents a novel approach to convex relaxation in ReLU neural networks by proposing multi-neuron relaxations. Specifically, it explores two types: a multi-neuron relaxation for the output of a layer and a multi-neuron relaxation for the functional graph. The authors aim to simplify verification by directly bounding the function value range, which could provide a more complete and efficient verification process. I appreciate the focus on finding complete verifiers, as achieving precise bounds for functions computed by neural networks is a challenging yet impactful goal. Furthermore, the case study on max functions suggests that multi-neuron relaxation could improve guarantees, potentially motivating further research in this area. This multi-neuron approach appears promising and could contribute to more efficient verification techniques applicable to a wider range of CPWL functions.

**Weaknesses:**

However, the paper’s main results appear unsupported.

***Incorrect Lemma*** Lemma 3 claims that ReLU neural networks transform polytopes into polytopes. Here is a simple counterexample: consider the polytope $P=[-1,1]$ and the CPWL function $f \colon R^1 \to R^2$ given by $x \mapsto (x,\max ${$x,0$}$)$ which can clearly be computed by a 2-layer neural network. Then, $f(P)$ is not a polytope since it is simply the graph of the ReLU function. The issue in the proof is that it concludes that applying the ReLU function to a polytope results in intersecting the polytope with the half-spaces $y \geq 0$. However, this is not the case; the ReLU function maps everything not in the positive orthant to its boundary, which may result in a nonconvex set.

Since the results for both relaxations $M_3$ and $M_{d_{in}+3}^o$ rely on this lemma, the paper’s main results appear unsupported.

***Lack of Precision***
 Aside from Lemma 3, the rest of the paper also lacks precision, which made it challenging for me to fully understand the remaining results and verify their correctness. For example:
- The definition of $M_{k}^o$ is unclear. The paper states: “Concretely, $C(x, \rho(x))$ is in the form of {$C(\rho(x_I )) | I \subseteq [d], |I| \leq k$}, and only one index set $I$ is allowed per layer as well." This set appears to range over all possible $I$, but it then states that only one index set $I$ is allowed, which seems contradictory. While I think I understand what is meant, a precise formal definition of the central objects of study would be desirable.
- Moreover, if I understood correctly, the relaxations depend on the set $I$ of output neurons chosen to compute the convex hull of its function values. This should be reflected in the notation to avoid ambiguity.
-  In the proof of Theorem 2, the justification for the second equality in the last equation is missing. Although it seems intuitively correct, it is challenging to verify formally given the lack of precision in the definition. Could you provide a more detailed proof of this step?

**Questions:**

- Is there a way to replace Lemma 3 by a weaker statement such that (parts) of you results still hold?
- Could you provide a time complexity analysis for computing the $M_k$ and $M_k^o$ relaxations in terms of the network size and the parameter $k$?
- What is the computational complexity of determining unstable and stable neurons in this context?

I encourage the authors to address the issues with Lemma 3, improve precision in definitions and proofs, and include a discussion on the runtime of computing the relaxations.

---

### Official Review · Reviewer_Cmy7 · 2024-11-03

**Soundness:** 3
**Presentation:** 3
**Contribution:** 2
**Rating:** 5
**Confidence:** 3

**Summary:**

This paper proves that multi-neuron relaxations enable an exact characterization of the output bound of a neural network, while the finest single-neuron relaxations fail to achieve. The paper considers two specific type of multi-neuron relaxations, namely the k-ReLU relaxation and the output-only k-ReLU relaxation. They prove that any piecewise linear function can be expressed by a ReLU function of which the output can be exactly bounded by an output-only $d_{in}+3$ - ReLU relaxation or a multi-neuron relaxation of order 3. Their proof follows by introducing the concept of unstable neurons, illustrating that the number of unstable neurons is what matters, and using a construction from [1] to show that there exists a representation of any piecewise linear function as a network of finite depth and having at most 3 unstable neurons. The paper discusses a particular example of the max function, and discussed the implications of the theorem, the computational complexity and usefulness for neural network certification.

[1] Hanin, Boris. "Universal function approximation by deep neural nets with bounded width and relu activations." Mathematics 7.10 (2019): 992.

**Strengths:**

1. The paper is well written, giving sufficient motivation for what problem the authors are trying to solve. Especially, I am convinced that this paper tries to tackle an important problem that people working on certification will be interested in their results.
2. The examples the authors provide illustrate what they want to say in a clear manner. Especially, section 7 illustrates why the result is important very well.
3. The novelty of the result is clear, and though some parts of the result builds on existing results, the ideas connecting stability of neurons and multi-neuron relaxation in Lemma 7 and 8 seems novel enough and written in a clear manner.

**Weaknesses:**

1. Something that I am not convinced about is the proof of Theorem 2 and Lemma 3. I think there is a flaw in the proof and we have a counterexample.

Flaw in the proof: In the induction step we assume that the case of $L = 2$ is proved. However, as we assume that all depth of $\leq L-1$ is proved, it should imply that $3 \leq L$. This means that the case of $L = 2$ should have been proved as a part of the base case, and a counterexample occurs here.

Counterexample: Let $L = k = d_{in} = d_{out} = 2$. Let $X = \{(x, y)\ |\ -2 \leq x+y \leq 2, \ -4\leq x-y \leq 4 \}$. Also $W_1 = W_2 = I$. Then, $f(x) = \rho(x)$ is a network ended with an affine layer. Also, $X$ is a convex polytope. However, $f(X)$ is given as

$$
f(X) = \{(x,y) \ | \ 0\geq x, 0 \geq y, x+y \leq 2\} \cup \{(0,x) \ | \ 0 \leq x \leq 4\} \cup \{(x,0) \ | \ 0 \leq x \leq 4\}.
$$

which is not convex. This also becomes a counterexample of Thm 2, because $\mathbb{M}^{o}_{k}(f,X)$ is convex.

Probably there is a misunderstanding in the proof or the statement of the theorem that I did not fully grasp. However, currently I believe this is the flaw of Thm 2 and Lemma 3, and as Thm 6 uses Thm 2, most parts of the proof should be fixed. Even so, section 7 is true, which I believe is an important result.

**Questions:**

1. One suggestion is that the authors could reorder the Theorems and definitionsso that they are ordered as the occurence. Some examples:

- Definition of stability of neurons in backgrounds?
- Put Lemma 1 in front of Theorem 1.
- Put Lemma 7 and 8 in front of Theorem 6.

2. I am curious: doesn't Theorem 2 imply Lemma 3 (as $\mathbb{M}^{o}_{k}(f, X)$ is convex)?

3. As general ReLU functions have exponentially many piecewise linear regions (exponential in the input dimension), and depth in Hanin's construction depends on the number of piecewise linear regions, I think Lemma 1 gives a network that is too deep. Is using Lemma 1 to apply in cases of practical certified training actually practical (as proposed in pg 10)?

4. If you could address the weakness, it would be great. If there is any misunderstanding or mistake that I have, please illustrate it and I would raise my score.

---

### Official Review · Reviewer_u8oc · 2024-11-04

**Soundness:** 3
**Presentation:** 2
**Contribution:** 3
**Rating:** 5
**Confidence:** 4

**Summary:**

This paper addresses the question of whether convex relaxations of ReLU neural networks can provide exact bounds for general continuous piecewise linear functions in $\mathbb{R}^n$. It has been previously established that single-neuron based relaxation methods cannot provide exact bounds even for a simple "max" function in $\mathbb{R}^2$. The results in this paper show that layer-wise multi-neuron relaxations can compute exact bounds for general ReLU networks. First, they establish the full-expressivity of multi-neuron relaxations under bounded width assumption and then with bounded unstable neurons.

**Strengths:**

The main strength of this paper is that it attempts to establish the full expressivity of ReLU neural networks under multi-neuron relaxations for the first-time.

**Weaknesses:**

The presentation of the results need a lot of work. The concepts and results are presented in a backward way with the main results presented first with a lot of unclear terminology and then the preliminaries are presented later. For instance, one needs to read till Section 6 to understand what an unstable neuron is, and the proof of the main result in Section 4 refers to Lemma 1 which states that there exists a ReLU network $\Phi$ of width $d_{in} + 3$ with at most 3 unstable neurons per layer. The word "completeness" is used in place of "full expressivity" without clear definition in many places. The proofs are not written very clearly and are not easy to follow. I would suggest creating a preliminaries section where the basic definitions like "stability" and "bounded stability" are clearly defined and basic lemmas like Lemma 1 are clearly stated with the proofs in the appendix. This should be augmented with Section 3 and done before Section 4 and Section 5. There is no experimental evaluation. For the experimental evaluation, one use single-neuron and multi-neuron relaxations with several different neural network architectures, ranging in width and depth, to show that the computed bounds are exact. This has potential to be a good paper but it needs better presentation of the results and more experimental evidence.

**Questions:**

1. For the proof of Theorem 2, the base case $L = 1$ the function $f(\mathbf{x}) = \mathbf{A}\mathbf{x} + \mathbf{b}$ is not a ReLU network. It is just a single affine layer without a ReLU activation following it.The base case should be $f(\mathbf{x}) = \rho(\mathbf{A}\mathbf{x} + \mathbf{b})$. Same goes for proof of Theorem 6.
2. In the proof of Lemma 7, you should replace "all effective $l_2(x, v_{1:k})$" with "all effective constraints $l_2(x, v_{1:k})$". Maybe better to use the term "active" instead of "effective" or you can add the sentence explaining what "effective" constraint means before you use the term.
3. In the proof of Lemma 7, what is the relationship between $l_1$ and $l_2$ combined and $l$?
4. In the proof of Lemma 8, you use $X_L$ and $\mathbb{X}_L$ as well as $Y_L$ and $\mathbb{Y}_L$ interchangeably. What is the difference between them?
5. You use $\mathbb{M}_\infty$ throughout without really explaining what it is physically. Does it correspond to the infinite-width case where there are infinite number of neurons in a layer and all the neurons are considered in the convex relaxation?

---

### Official Review · Reviewer_A22N · 2024-11-09

**Soundness:** 3
**Presentation:** 4
**Contribution:** 2
**Rating:** 5
**Confidence:** 3

**Summary:**

The authors use multi-neuron relaxations to prove the existence of exact convex relaxations of arbitrary ReLU network images under convex polytope input sets. In turn, this proves the existence of exact convex relaxation for arbitrary piecewise continuous functions with a quantitative statement about the network size.

**Strengths:**

- This result is of theoretical interest to the neural-network robustness / verification community

- The paper is well-written and the technical results are presented in a clear and accessible manner. Although I did not check the proofs carefully.

- The authors include a nice example of applying their multi-neuron relaxation to the motivating example of their paper: the max function in $\mathbb{R}^2$.

**Weaknesses:**

- The area of neural network certification is at least motivated by the practical problem of proving robustness of a network's output to the specified perturbation,  in particular in classification (Szegedy 2014). The lack of implementable algorithms or even complexity of computing such a relaxation is concerning. There are some appealing points made in the discussion of section 8 such as avoiding BaB procedures, but the impact of this paper would be much greater if more details were provided towards an implementation. Without this, I'm left unsure about the concrete impact for the ICLR community or it may be outside of my own current scope.

**Questions:**

- It might be helpful to include some basic definitions and background earlier in the paper. The notion of stable and unstable neurons is not defined until page 6, but mentioned frequently before that.

- As mentioned in the paper there already exists automatic and complete verifiers such as alpha-beta CROWN [Wang 2021] which employs a combination of IBP and BaB. Although BaB can get quite expensive with higher dimensions, it results in a complete verifier which can be implemented on multiple GPUs. It might be helpful to give more concrete context about how your relaxation could be utilized and advance state-of-the-art.

Minor comments:
- In the very first sentence of the abstract: “Neural work certification” -> “Neural network certification”

[Wang 2021] Beta-CROWN: Efficient Bound Propagation with Per-neuron Split Constraints for Neural Network Robustness Verification

---

### Note · Authors · 2024-11-19

**Comment:**

We would like to thank everyone for their efforts in improving this paper. In particular, Reviewer Cmy7 and 6s15 give different constructions on a counterexample of Lemma 3. The key problem happens in Line 269, where the range of a ReLU function is mistakenly presented. Therefore, we feel necessary to withdraw this work for now, and submit again after we fix the proof.

**Withdrawal Confirmation:**

I have read and agree with the venue's withdrawal policy on behalf of myself and my co-authors.